# Real-Time Grading of Defect Apples Using Semantic Segmentation Combination with a Pruned YOLO V4 Network

**DOI:** 10.3390/foods11193150

**Published:** 2022-10-10

**Authors:** Xiaoting Liang, Xueying Jia, Wenqian Huang, Xin He, Lianjie Li, Shuxiang Fan, Jiangbo Li, Chunjiang Zhao, Chi Zhang

**Affiliations:** 1Intelligent Equipment Research Center, Beijing Academy of Agriculture and Forestry Sciences, Beijing 100097, China; 2College of Information Technology, Shanghai Ocean University, Shanghai 201306, China; 3National Research Center of Intelligent Equipment for Agriculture, Beijing 100097, China

**Keywords:** defective apples, apple grading, deep learning, object detection, semantic segmentation

## Abstract

At present, the apple grading system usually conveys apples by a belt or rollers. This usually leads to low hardness or expensive fruits being bruised, resulting in economic losses. In order to realize real-time detection and classification of high-quality apples, separate fruit trays were designed to convey apples and used to prevent apples from being bruised during image acquisition. A semantic segmentation method based on the BiSeNet V2 deep learning network was proposed to segment the defective parts of defective apples. BiSeNet V2 for apple defect detection obtained a slightly better result in MPA with a value of 99.66%, which was 0.14 and 0.19 percentage points higher than DAnet and Unet, respectively. A model pruning method was used to optimize the structure of the YOLO V4 network. The detection accuracy of defect regions in apple images was further improved by the pruned YOLO V4 network. Then, a surface mapping method between the defect area in apple images and the actual defect area was proposed to accurately calculate the defect area. Finally, apples on separate fruit trays were sorted according to the number and area of defects in the apple images. The experimental results showed that the average accuracy of apple classification was 92.42%, and the F1 score was 94.31. In commercial separate fruit tray grading and sorting machines, it has great application potential.

## 1. Introduction

Recently, consumers’ awareness of fresh fruit quality is increasing. They often prefer to buy apples with regular shapes, smooth surfaces and no obvious scars and damages. Therefore, it is particularly important to detect and grade apples before they are sent to the market, which will greatly improve the income of fruit farmers. Apples can be classified into different grades based on basic characteristics such as size, shape, color and whether they are defective. However, it is still a challenging task to accurately detect the apple defect area on the automatic sorting line. Especially for expensive fruits, if the number and area of defects are not considered and all defective apples are treated as substandard fruits, it will cause potential economic losses to fruit farmers. Therefore, the detection and grading of apple surface defects is an urgent problem to be solved for expensive apple grading and sorting.

Mizushima et al. [1] applied a linear support vector machine (SVM) and Otsu method to classify apples. First, the optimal classification hyperplane was calculated, and then the color image was gray scaled with SVM. The optimal threshold near the fruit boundary was obtained by the Otsu method. Finally, apples were eventually divided into three commercial grades. Jawale et al. [2] proposed the K-means clustering method to segment the image. Then, an artificial neural network (ANN) combined with color and texture features was used to separate the defective apples. Mohammadi et al. [3] used a simple threshold to extract gray-scale images. Then, the shape features such as roundness of the segmented apples were obtained and used to detect defects. Nosseir [4] proposed an algorithm to identify rotten fruit by extracting RGB value. The recognition accuracy of the method was 96.00%. Wenzhuo Zhang et al. [5] proposed an apple suspicious defect detection method based on a multivariable image analysis strategy. The FCM-NPGA algorithm was used to segment the suspicious apple defect image. The overall detection accuracy was 98%. Chi Zhang et al. [6] used NIR-coded structured light and fast lightness correction to automatically detect defective apples. Defective regions or stem/calyx regions can be correctly distinguished. The identification rate of defective apples with this method was 90.2%. Integrating the four characteristics of apple size, color, shape and surface defects, the apples were divided into three levels by support vector machine (SVM). The detection accuracy of surface defects based on a single index was 95.85%, while the average classification accuracy of apple surface defects based on multiple features was 95.49%.

The feature extraction method is the key to the accuracy of fruit detection for traditional machine vision technology. However, some methods require complex acquisition systems, and some may not be used in online applications [7]. Recently, multispectral imaging (MSI) and hyperspectral imaging (HSI) systems have been applied to nondestructive detection of fruits such as apples, oranges, etc. [3,8]. However, due to the time consumption of image acquisition and the high price of the HSI camera, the practical application of HSI was limited [9,10]. Huang et al. [11] used principal component analysis (PCA) to detect apple defects in hyperspectral images. However, the classification accuracy in the online test was 74.6%. In addition, the research based on a laboratory MSI system could only deal with defect detection under static conditions, which was difficult to apply to online detection.

Due to the fast detection speed and low cost of the RGB color camera, traditional machine vision using an RGB color camera had obvious advantages in online fruit grading and sorting based on color, size, shape and defect compared with other nondestructive testing technologies [12]. In recent years, deep learning has been widely used in agriculture, industry, medicine and other fields. It automatically learned image features from the input image and the key features with fewer human factors were extracted for subsequent tasks [13]. Machine vision based on an RGB color camera combined with various deep learning models greatly improved the online grading accuracy of apples.

For postharvest quality grading of ordinary apples, it is sufficient to divide apples into normal and defective apples without locating the defects of each apple [10]. Therefore, Yujian Xin et al. [14] compared the detection results of SVM, Fast RCNN, YOLOv2 and YOLOv3 models on apple images. The YOLOv3 model had the best effect on apple defect detection. The average detection time of an apple image was 1.12 s, and the F1 score was 92.35%. Paolo et al. [15] regarded apple defects as object detection problems. After comparing a single shot detector (SSD) with YOLOv3, the YOLOv3 model was trained using a dataset containing healthy and defective apples to detect which apples were healthy. The overall mAP was less than 74%. Guangrui Hu et al. [16] used the TensorFlow deep learning framework and SSD deep learning algorithm to identify apple surface defects. Yanfei Li et al. [17] proposed a fast classification model of apple quality based on a convolutional neural network (CNN) and compared it with the Google InceptionV3 model and HOG/GLCM + SVM. It was concluded that the accuracy of apple quality classification was 95.33%. Fan et al. [18] compressed the depth and width of the YOLO V4 network through channel pruning and layer pruning, which reduced the inference time and model size of the network by 10.82 ms and 241.24 MB, respectively, and increased the mAP to 93.74%. This method was suitable for the defect identification of different varieties of apples. Zhipeng Wang et al. [7] proposed an object-detection algorithm based on YOLOv5. The real-time detection of apple stem/calyx could be realized, and the detection accuracy was 93.89%.

However, current research on apple surface defect detection has either been on the condition of a static environment or based on online detection using a roller conveyor. Although sorting machines with a roller conveyor have fast sorting speed, it is easy to cause mechanical damage to the apple and reduces the quality of the apple when the apples are rotated with the roller. The widely used roller conveyor sorting equipment requires that the fruit to be sorted has high hardness. Although image processing techniques are applied to sorting fruits, sorting fruits according to the number and area of surface defects is still a difficult problem. For fragile fruits with low hardness, it is easy to cause damage and economic losses when sorting with chain transmission equipment. Fruits with higher prices are also likely to cause potential economic losses in the process of sorting with a roller conveyor. Therefore, a fruit sorting machine based on separate fruit trays was designed which could protect the apples from damage. The separate fruit tray has been especially suitable for online grading of high-quality apples. With the improvement of classification requirements, it was not only necessary to determine whether the apple’s surface had defects but also to identify the number and area of apple surface defects. For expensive high-end fruits, if both slight defects and severe defects were considered as equal defective fruits and discarded, this would cause economic losses to farmers, so it is necessary to grade fruits according to the number and size of defects. Due to the curvature of the fruit’s shape, the area of defect in the image would be compacted compared with actual defect area. Therefore, it was necessary to further accurately grade the defective apples in high-quality apples according to the number and area of defects, so as to reduce the economic losses of fruit farmers.

In this paper, a defect grading method based on deep learning is proposed to identify the number and area of defects in an apple image. The specific objectives were: (1) using the BiSeNet V2 network to build a defect detection and segmentation model, (2) using the YOLO V4 network to correct the results of BiSeNet V2 detection, (3) building the corresponding relationship between the number of pixels in the defect area of an apple image and the actual defect area, and (4) grading the defective apple according to the defect area and quantity.

## 2. Materials and Methods

### 2.1. Samples

The samples were composed of 180 defective Fuji apples and 50 healthy Fuji apples. Apples with different degrees of defects and healthy apples were picked in November 2021 from a commercial orchard in Beijing. Before capturing apple images, mud points on the apples’ surface were washed to avoid mistaking them as defects.

### 2.2. Computer Vision System

The computer vision system was composed of industrial control computer, RGB camera (acA1920-40 gc, Basler, German), lens (M0814-MP2 8 mm, Computar, Japan), hemispherical lighting hood and the outermost light chamber (Figure 1). There was a circular opening at the top of the hemispherical lighting hood and a circle of light-emitting diode (LED) lights at the bottom of the hemispherical lighting hood. All components (except industrial control computer) were fixed in the light chamber.

The quality of apple images was directly related to the detection accuracy of apple defects. It was significant to capture an image without any light spots. Direct illumination would bring about obvious bright spots on the apple. At the same time, the central regions of the apple images were bright, and the surrounding regions were dark, which increased the difficulty of accurate detection. Image quality, related to the performance of the illumination system, would affect the detection results of apple defects. It was quite important to adopt a suitable illumination system. Therefore, a hemispherical lighting hood with LED light source (wavelength range between 500 nm and 630 nm) was applied to realize the irradiation effect of diffuse reflection in this study.

The RGB camera (1920 × 1200 pixels) was installed directly above the hemispherical hood. The apple on the separate fruit tray continuously transmitted under the camera. Apple images could be captured through the circular opening at the top of the hemispherical lighting hood by the camera. In total, 112 LED lamp beads were built at the bottom of the hemispherical lighting hood to form a circular light source. The power of a single LED lamp bead is 3 W, and the color temperature is 6500 k. The LED lights were controlled by the hardware trigger in the control unit. The output voltage of LED power supply is continuously adjustable from 13 V to 24 V, and the light intensity can be adjusted by manually rotating the button of LED power control unit. When an apple passed, the LED lights were on and off for the rest of the time. White diffuse reflective coating was painted on the inner surface of the lighting hood and the reflectivity was 99%, which could obtain uniform illumination. Therefore, the apple images in this study do not need corrected brightness.

The following frameworks were used to obtain the segmentation model and detection model, respectively, in this study: PaddleSeg-based framework (Baidu, China) of version 2.1 for BiSeNet V2 and Darknet-based framework (open-source framework) for YOLO V4, together with Python version 3.7. All experiments were performed on a 64 bits Intel Core i7-6700 CPU with 3.4 GHz and 32 GB RAM memory. One graphics processing unit (GPU), GeForce GTX 2080 with 8 GB of memory under CUDA version 10.1, was employed in this study. The operating system was Windows version 10. C++ language was used to realize online deployment.

### 2.3. Image Dataset

The apple images used in this research were captured by the machine vision system, as shown in Figure 1. Before capturing the apple images, the apples were put on the separate fruit tray, and the separate fruit tray moved with the conveyor belt. When the apples passed through the lighting chamber, the camera on the top of the lighting chamber would automatically capture the apple image directly under the control of hardware trigger signal. Then, grading software read the apple image from the camera buffer and saved the image. Three thousand apple images were finally obtained as the dataset of this research. The size of all apple images was 400 pixels × 336 pixels. Before training, the resize function of openCV was used to resize the input images into 512 × 512 pixels.

An open-source annotation tool-LabelMe-was used to semantically label the captured apple defect images and establish a standard semantic label dataset. Meanwhile, LabelImg was used to mark the stem, calyx and defect regions in the apple images. In total, 2400 images were selected as the training set of the network and the remaining 600 as the validation set.

### 2.4. Apple Surface Defect Detection Based on BiSeNet V2

In order to obtain an optimal lightweight network model to reduce the network parameters, many researchers were looking for a balance among the amount of computation, parameters and accuracy, hoping to use as few computations and parameters as possible to obtain high accuracy of the detection model [19]. In the field of semantic segmentation, reducing the image size or reducing the complexity of the model could decrease the computation cost caused by semantic segmentation.

Reducing the image size could directly reduce the amount of computation, but the image would lose many details, which would affect the image accuracy. In addition, reducing the complexity of the model would weaken the feature extraction ability of the model, which would affect the segmentation accuracy. Therefore, it was quite challenging to apply lightweight model in semantic segmentation task while taking into account accuracy and real-time performance.

The BiSeNet network could basically balance the relationship between real-time performance and accuracy [20]. So, it was used in this research, and the architecture of it is shown in Figure 2.

The BiSeNetV2 network is divided into three main components: the two-pathway backbone (green dashed box) with a detail branch (the purple cubes), a semantics branch (the pink cubes), the booster component (blue dashed box) and the aggregation layer (red dashed box). C1, C2 and C3 indicate the channels of the detail branch, respectively. The context embedding block as the output of the semantics branch is in the last stage. Down and up represent the down-sampling and the up-sampling operation, respectively. The sigmoid function and the elementwise product were represented by φ and ⊗, respectively.

Shallow layers and wide channel dimensions are the characteristics of the detail branch, which have a small receptive field of spatial detail used to generate high-resolution feature representation and capture low-level detail. The semantic branch with deep layers and narrow channel dimensions has a large receptive field for the categorical semantics to capture high-level semantics. The gaps between the semantic and resolution were compensated by the aggregation layer. The initialization parameters of BiSeNet V2 network are shown in Table 1.

Because defects were considered as the region of interest in apple images and in order to ensure the real-time detection, apple images were only segmented into defect region and background region. The segmentation result based on BiSeNetV2 used binary image I_B_ to present. The gray value of defect region was set as B_V_, where B_V_ was not equal to 0. The gray value of background region was set as 0. In practical application, there might be multiple defect regions in apple images. So, R_B_ (R_B_∈{R_b1_, R_b2_,... R_bn_}) was used to store the position values of different defect regions, where *n* was the total number of defects obtained using BiSeNetV2 model in apple image.

The overall goal of this study was to quickly and accurately realize the online grading of defective apples. Therefore, it was necessary to further calculate the area and the number of defects of defective apples. Finally, the grade of apple could be determined according to the comparison between the defect information and the grading standard.

### 2.5. The Correction of Apple Defect Detection Based on Pruned YOLO V4 Network

Although as a lightweight semantic segmentation model BiSeNet V2 could realize real-time semantic segmentation, it might incorrectly segment the apple stem/calyx region as defect region. Therefore, the object detection model was further used to accurately determine the location of the defect region.

The result of object-detection algorithm required not only identifying the object category in the pictures but also marking the position parameters of the objects. Among them, RCNN, Fast RCNN, SPP-Net [21] and Faster RCNN [22] could be divided into two main parts: region proposal and extraction regions. Therefore, YOLO model had less computation and was faster than two parts methods, as YOLO model replaced numerous regions through grid division and anchor method. YOLO V4 [23] model was implemented based on Darknet framework, which could easily and flexibly use C++ language to deploy the trained network model in practical application. Therefore, a defect detection model based on the YOLO V4 was proposed to identify the defect region in RGB apple images.

The YOLO V4 object-detection algorithm was an improved version of YOLO V3 [24]. Compared with the YOLO V3 object-detection algorithm, YOLO V4 improved the speed and accuracy of real-time detection of the algorithm [25].

CSP (Cross Stage Partial) module could improve the learning ability of the network. CBL module was composed of the Convolution, batch normalization and Leaky_ReLU and CBM module was composed of the Convolution, batch normalization and Mish [26]. These two modules were used to extract input image features. SPP (Spatial pyramid pooling) module used the max-pooling of different scales to pool the input feature layers and then stacked them, which could greatly increase the receptive field.

Recently, YOLO V4 has been used for defect detection of a variety of objects. With the proposal of YOLOv5 and YOLOX, many researchers focus on the newly proposed network, but from the perspective of practical application, YOLO V4 is easier to deploy and realize the online detection of apple defects. The architecture of YOLO V4 network is shown in the Figure 3.

In order to realize the real-time detection of apple surface defects, it was necessary to deploy the YOLO V4 model into the apple automatic grading system. After experiment comparison, it was found that a large number of network parameters, network layers and complex structure of YOLO V4 lead to excessive calculation. It could not meet the real-time requirements of apple surface defect detection. Considering the memory occupation of the YOLO V4 and ensuring the real-time and stability in the detection process, a lightweight model, pruned YOLO V4, was obtained by compressing the YOLO V4 network based on RGB images of apples. Model pruning method could achieve the best balance between model detection speed and detection accuracy. It was also a method to automatically obtain the simplest network structure of the original model. The pruned YOLO V4 model could be deployed on the Windows 10 operating system and hardware and realized the real-time detection of apple surface defects.

In order to obtain pruned YOLO V4, firstly, the sparsity training was introduced into the YOLO V4 network. The scale factors were sorted after sparsity training, and the maximum scale factor meeting the requirements of pruning rate was set as the threshold. Then, it deleted the channels that were less than the threshold. High contribution channels were retained, and low contribution channels were deleted according to the scale factor. So, the channel pruning was completed. However, the object detection model still could not meet the requirements of real-time detection after the channel pruning of the model, which compressed the network width. So, the depth of the network model was further compressed using the layer-pruning method. The mean value of the scale factor of each layer was sorted. The layer with the lower mean value was selected for pruning, which completed the layer pruning. After completing the channel pruning and layer pruning of the YOLO V4 model, the accuracy of the model may decline. Fine-tune operation was used to improve the detection accuracy of the pruned model. Finally, a pruned YOLO V4 model for defect detection was obtained. The result of pruning is shown in Figure 4.

Pruned YOLO V4 network could accurately locate the location of apple defects and identify the apple stem/calyx. For the defection of regions identified by the pruned YOLO V4 network, the corresponding location information of each defect region was compared with the binary image I_B_ generated by BiSeNet V2 network. For the position where the gray value Bv was in the defect area determined by pruned YOLO V4 network, it was determined as the defect area. Finally, the defect region in apple image was segmented accurately by combining BiSeNet V2 network and pruned YOLO V4 network.

### 2.6. Defect Area Correction

If the extent of surface defects of expensive fruits is not graded, all fruits containing defects could be sorted as substandard fruits, which would cause serious economic losses to fruit farmers. So, some grading standards of fruits classify fruits according to extent of defects. For example, the number and area of apple surface defects under different grades were restricted in the local standard for apple grading in Beijing, China. Therefore, the number and area of the defects in defective apples needed to be accurately calculated.

The surface of the apple has a certain curvature because of the similarity between an apple and a sphere. When the apple was placed on the separate fruit tray and the industrial camera captured image of the apple, the defects in different areas of the outer surface of the apple would be scaled to varying degrees. Therefore, the actual area of defect might be different from the region of defect in apple images. Thus, projection method was presented to provide a solution for building the relationship between actual area of defect and defect region in apple images.

In order to eliminate the influence of surface curvature on the defect area in the apple image, it was necessary to correct the number of pixels in the defect area of the apple image. Firstly, apple models of different sizes were obtained by 3D printing, referring to the characteristics of apple surface changes in orchards. There were 12 apple models with horizontal diameter from 68 mm to 90 mm, as shown in Figure 5. In order to establish the corresponding relationship between the number of real pixels and defective pixels in the apple image, a series of black square labels with a side length of 3 mm was printed and pasted on the surface of apple models of different sizes to simulate the change of defect area at different positions.

Then, each apple model was put under the camera and the images of the apple models were captured statically. The actual pixel value of the squares and their pixel value in the image were different because of the change of curvature and the different distance from each square to the center of the apple. Therefore, it was necessary to establish the function relationship between the three variables, namely the number Z of real pixels, the distance d from the defect region to the center of the apple and the r representing the radius of the apple. The function relationship could be represented as Z=F(d,r). In order to obtain the expression of the function, the number of pixels in the defect area of apple model image at different positions was recorded manually for apple models with different sizes. Then, the dataset (*d_w_*, *r_w_*, *z_w_*) corresponding to the three variables was generated.

So, given a dataset (*d_w_*, *r_w_*, *z_w_*), *w* = 1, 2, 3,..., *n*. The bivariate polynomial function F(d,r) based on the dataset could be expressed as:(1)$$\;F\left(d,r\right)=\overset{p,q}{\underset{\mathrm{ij}=1,1}{\sum }}g_{\mathrm{ij}}d^{i-1}r^{j-1}=\overset{p}{\underset{i=1}{\sum }}\overset{q}{\underset{j=1}{\sum }}g_{\mathrm{ij}}d^{i-1}r^{j-1}$$

Let
(2)$$d=\left[\begin{array}{c} 1\\ d\\ d^{2}\\ \vdots \\ d^{p} \end{array}\right],r=\left[\begin{array}{c} 1\\ r\\ r^{2}\\ \vdots \\ r^{p} \end{array}\right],G=\left[\begin{array}{ccc} g_{11} & \cdots & g_{1q}\\ \vdots & \ddots & \vdots \\ g_{p1} & \cdots & g_{\mathrm{pq}} \end{array}\right]$$

Then, the function could be expressed as
(3)$$\;F\left(d,r\right){=d}^{T}\mathrm{Gr}$$

The goal of fitting was to obtain the parameter matrix G. To obtain the parameter matrix G, a multivariate function with respect to the parameter *g_ij_* was constructed:(4)$$\;L\left(g_{11},\cdots {,g}_{\mathrm{pq}}\right)=\overset{n}{\underset{w=1}{\sum }}{\left[F\left(d_{w}{,r}_{w}\right){-z}_{w}\right]}^{2}=\overset{n}{\underset{w=1}{\sum }}{\left(\overset{p}{\underset{i=1}{\sum }}\overset{q}{\underset{j=1}{\sum }}g_{\mathrm{ij}}d^{i-1}r^{j-1}{-z}_{w}\right)}^{2}$$

The point (*g*_*11*_,..., *g_pq_*) was the minimum point of the multivariate function L (*g_11_*,..., *g_pq_*), and *z_w_* was the number of actual pixels, so the point (*g_11_*,..., *g_pq_*) must satisfy the equation:(5)$$\frac{\partial L}{\partial g_{\mathrm{ij}}}=2\overset{n}{\underset{w=1}{\sum }}\left[d_{w}^{i-1}r_{w}^{j-1}F\left(d_{w}{,r}_{w}\right){-d}_{w}^{i-1}r_{w}^{j-1}z_{w}\right]=0$$

So, the following equation could be obtained:(6)$$\overset{n}{\underset{w=1}{\sum }}d_{w}^{i-1}r_{w}^{j-1}F\left(d_{w}{,r}_{w}\right)=\overset{n}{\underset{w=1}{\sum }}d_{w}^{i-1}r_{w}^{j-1}z_{w}$$

According to Equation (1), there were:(7)$$\overset{n}{\underset{w=1}{\sum }}d_{w}^{i-1}r_{w}^{j-1}z_{w}\overset{p}{\underset{\alpha =1}{\sum }}\overset{q}{\underset{\beta =1}{\sum }}g_{\alpha \beta }d_{w}^{\alpha -1}r_{w}^{\beta -1}=\overset{n}{\underset{w=1}{\sum }}d_{w}^{i-1}r_{w}^{j-1}z_{w}$$
(8)$$\overset{p,q}{\underset{\alpha \beta =1,1}{\sum }}\left[g_{\alpha \beta }\overset{n}{\underset{w=1}{\sum }}\left(d_{w}^{\alpha -1}r_{w}^{\beta -1}d_{w}^{i-1}r_{w}^{j-1}\right)\right]d_{w}^{i-1}r_{w}^{j-1}z_{w}=\overset{n}{\underset{w=1}{\sum }}d_{w}^{i-1}r_{w}^{j-1}z_{w}$$

Let $u_{\alpha \beta }(i,j)=\overset{n}{\underset{w=1}{\sum }}{(d}_{w}^{\alpha -1}r_{w}^{\beta -1}d_{w}^{i-1}r_{w}^{j-1})$ and $v(i,j)=\overset{n}{\underset{w=1}{\sum }}d_{w}^{i-1}r_{w}^{j-1}z_{w}$

So, Equation (8) can be rewritten in matrix form:(9)$$\left[\begin{array}{ccc} u_{11}\left(1,1\right) & \cdots & u_{\mathrm{pq}}\left(1,1\right)\\ \vdots & \ddots & \vdots \\ u_{11}\left(p,q\right) & \cdots & u_{\mathrm{pq}}\left(p,q\right) \end{array}\right]\left[\begin{array}{c} g_{11}\\ \vdots \\ g_{\mathrm{pq}} \end{array}\right]=\left[\begin{array}{c} v(1,1)\\ \vdots \\ v(p,q) \end{array}\right]$$

Equation (9) could be rewritten as the form Ug = V, where U is matrix with *pq* × *pq*, and V is a column vector with length *pq*. The column vector g could be calculated. Then, g was transformed into the parameter matrix G. So, the function F(d,r) could be determined using matrix G.

The object distance between the apple and the camera lens would change due to the different size of the apple, and it would affect the conversion between the real pixels and real area corresponding to the real pixels. In order to determine the corresponding relationship between the number of defective pixels and the actual defect areas under different apple sizes, every model apple with a certain size was used to determine the calibration coefficient c(*r*) between the number of pixels and the real areas S, where *r* was the radius of apple. For c(*r*)∈C, C was composed of 20 calibration coefficients. The final area projection equation was:(10)$$S=c(r)\underset{i\in S_{I}}{\sum }{F(d}_{i},r)$$
where *i* represents the pixel located in the defect region *S*_I_ in the apple image.

According to Equation (1) to Equation (9), the actual number of pixels corresponding to the defect in the image could be determined. Then, the actual area of the defect could be obtained according to Equation (10). So, the grade of defective apple could be determined according to the apple grading standard.

### 2.7. Evaluation Metrics of the Model Performance

Several indicators [27] were used to evaluate the performance of the proposed model, such as accuracy (A), pixel accuracy (PA), mean intersection over union (MIoU), mean pixel accuracy (MPA), recall (R), precision (P) and F1 value, where TP, FP, TN and FN represented true positive, false positive, true negative and false negative, respectively.
(11)$$A=\frac{\mathrm{TP}+\mathrm{TN}}{\mathrm{TP}+\mathrm{TN}+\mathrm{FP}+\mathrm{FN}}$$
(12)$$\mathrm{PA}=\frac{\sum_{i=0}^{k}p_{\mathrm{ii}}}{\sum_{i=0}^{k}\sum_{j=0}^{k}p_{\mathrm{ij}}}$$
(13)$$\mathrm{MPA}=\frac{1}{k+1}\overset{k}{\underset{i=0}{\sum }}\frac{p_{\mathrm{ii}}}{\sum_{j=0}^{k}p_{\mathrm{ij}}}$$
(14)$$\mathrm{MIoU}=\frac{1}{k+1}\overset{k}{\underset{i=0}{\sum }}\frac{p_{\mathrm{ii}}}{\sum_{j=0}^{k}p_{\mathrm{ij}}+\sum_{j=0}^{k}p_{\mathrm{ji}}{-p}_{\mathrm{ii}}}$$
(15)$$R=\frac{\mathrm{TP}}{\mathrm{TP}+\mathrm{FN}}$$
(16)$$P=\frac{\mathrm{TP}}{\mathrm{TP}+\mathrm{FP}}$$
(17)$$F1=\frac{2P\cdot R}{P+R}$$

Assume that there were *k* + 1 classes (0... *k*) in the dataset and 0 usually represented the background. *p_ij_* indicated that it was originally class *i* and was predicted to be class *j*, and *p_ji_* indicated that it was originally class *j* but was predicted to be class *i*. Pixel accuracy (PA) refers to the proportion of pixels predicted correctly in the total pixels. Mean pixel accuracy (MPA) was an improvement on PA. It calculated PA for each class and then averaged PA for all classes.

## 3. Results and Discussion

In order to quickly and accurately realize the online classification of defective apples, the number and the area of defects needed to be calculated after apple defects were detected. Therefore, three semantic segmentation methods including DAnet [28], Unet [29] and BiSeNet V2 were compared. The detection results of the semantic segmentation for comparison are shown in Figure 6. In Figure 6, the green mark was used to label the pixels of the defect area detected by the semantic segmentation methods. Using the DAnet and Unet networks, the stem/calyx region was more likely to be wrongly segmented as a defective region, while the BiSeNet V2 network had a higher segmentation accuracy than other networks.

The performance comparison of different semantic segmentation models is shown in Table 2. It was observed in the results presented in Table 2 that the mean pixel accuracy (MPA) of the three semantic segmentation methods for apple defect detection were up to 99%. BiSeNet V2 for apple defect detection obtained a slightly better result in MPA with a value of 99.66%, which was 0.14 and 0.19 percentage points higher than DAnet and Unet, respectively. In addition, the mean intersection over union (MIoU) of the semantic segmentation method based on BiSeNet V2 for apple defect detection was 80.46%, which was 6.38 and 6.53 percentage points higher than DAnet and Unet, respectively. The results showed that BiSeNet V2 had a better ability to identify apple surface defects that DAnet and Unet failed to identify. DAnet, Unet and BiSeNet V2 took 37.40 ms, 22.64 ms and 9.00 ms, respectively, for a single image. Inference time is an important factor in evaluating online detection models. BiSeNet V2 took the shortest time, which was 75.94% and 60.25%, shorter than DAnet and Unet, respectively. Meanwhile, BiSeNet V2 had a smaller model size than other models. After comparing the pixel accuracy, inference time, parameter quantity and model size of the models, BiSeNet V2 could give consideration to higher segmentation accuracy and real-time performance. Therefore, the BiSeNet V2 model could meet the actual requirement of apple defect online detection.

### 3.1. Analysis of Improvement Using Pruned YOLO V4

In the above discussion, the BiSeNet V2 network exhibits better accuracy and faster detection speed, but in practical applications, there is still mis-segmentation as shown in Figure 7. The pixels in stem/calyx regions were mistakenly identified as defects, respectively, using BiSeNet V2. In Figure 7, the green mark in the first row shows the defective parts. The pruned YOLO V4 network with higher accuracy could be used to solve this problem. The pruned YOLO V4 model was used to process the apple images after semantic segmentation. The detection results are shown in the second row of Figure 7. A green bounding box was used to label defect regions. A purple bounding box and yellow bounding box were used to label calyx and stem regions, respectively. The apple stem/calyx region and defect region in the images of the second row of Figure 7 were identified accurately. Finally, comparing the result of semantic segmentation with the result of object detection, the defect area confirmed by the two results at the same time was determined as the true defect area as shown in the last row of Figure 7. Finally, by combining BiSeNet V2 and the pruned YOLO V4 network, the defect region in apple images was obtained accurately. Therefore, the combination of BiSeNet V2 and YOLO V4 could improve the segmentation results of defect regions in apple images.

### 3.2. Results Analysis of Defect Area Calculation

According to the method of defect area correction, the calculation of apple defect area was tested. Apples with defects of different sizes in different regions were tested, respectively. Each defect was tested ten times. Then, the average value of each defect was calculated.

When the defect was located in region A, as shown in Figure 8, the defect area calculated according to the method of defect area correction was compared with the actual defect area. Three apples with defects of different sizes in region A were selected, and the defect areas on each apple were different. The actual number of pixels was calculated in the defect region according to Equation (1). Then, the area of the defect region was computed according to Equation (10). Finally, the result was compared with the actual defect area. The measurement process was repeated ten times for each apple. Then, the average value of each area was calculated. Similarly, the calculated results of three apples with defects of different sizes in region B and region C are shown in Figure 9. It could be concluded that whether the defect was located in region A, region B or region C, the difference between the calculated defect area and the actual defect area were less than 2.23 mm^2^. When the defect was located in region C, the mean square error between the calculated area and the actual area was between 3.03 and 3.22, which was higher than that of defects in region A and region B. It could be concluded from Table 3 that when the defect area was large, the error of the defect area obtained was also large. This was mainly because when the samples used for training the semantic segmentation model were marked manually, it was difficult to accurately mark the edge of the apple defect region. Therefore, there might be a certain transition area between the defect edge region and the normal peel region, which led to the error in the calculated result of the defect area. Meanwhile, the sensor of the industrial camera used in the machine vision system was a CMOS chip, which led to differences in the images captured every time, even in the same conditions. This further leads to the error of defect area calculation.

### 3.3. Results of the Defective Apple Grading

In order to verify the detection effect of the proposed method of defective apple grading, 68 first-class apples, 64 second-class apples and 62 third-class apples were purchased in a supermarket and selected for testing referring to the grading standard of apples in Beijing. The experimental results are shown in Table 4 and Figure 10.

As shown in Table 4, the precision and recall in the three grades of apples were above 90.63%, and the overall precision and recall were 94.30% and 94.33%, respectively. The detection accuracy of apples was 92.42%, and the F1 value was 94.31%. Among the three grades, the apple grade identification with the highest precision was the first-class apples (95.59%), and the ones with lowest precision were second-class apples (92.06%). Due to the error that would occur when the area values of defect were close to the junction of two adjacent grades, first class and third class might be misclassified into second class, and second-class could also be misclassified into first class and third class. Meanwhile, if the defect was located at the edge of apple, it was sometimes incorrectly detected as the background by the object-detection algorithm and semantic segmentation model, which would also lead to the reduction of classification accuracy.

## 4. Conclusions

In this paper, a grading method of defective apples was proposed and applied to the separate fruit tray sorting machine. The BiSeNet V2 network and pruned YOLO V4 network were combined to extract the defect regions in apple images. The BiSeNet V2 network was utilized to determine the latent location of defect regions. The pruned YOLO V4 network was used to remove the non-defective region. A projection algorithm was proposed to build the corresponding relationship between the defect area in the image and the actual defect area on the apple’s surface. After the two deep learning models were deployed using C++ language, the average accuracy and the F1 score of defective apple grading in the online test were 92.42% and 94.31%, respectively.

The overall results denoted that the proposed method has potential to be implemented in commercial fruit-grading machines. Meanwhile, the proposed method has the potential for being extended to other fruit. Because separate fruit tray grading equipment in the market can only capture the upper surface of the fruit, we are developing a flexible air suction device to assist the camera with capturing the full surface image of the fruit. Future work will focus on improving the segmentation accuracy of defects and the projection accuracy of the defect area for improving the accuracy of grading defective apples.

## Figures and Tables

**Figure 1 foods-11-03150-f001:**
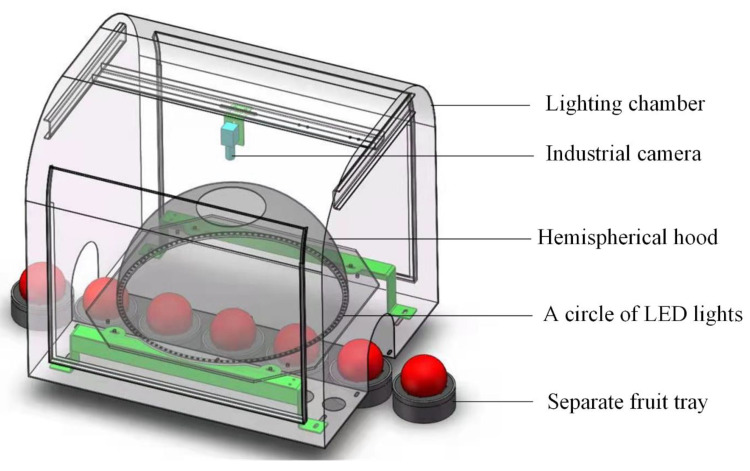
Image acquisition system.

**Figure 2 foods-11-03150-f002:**
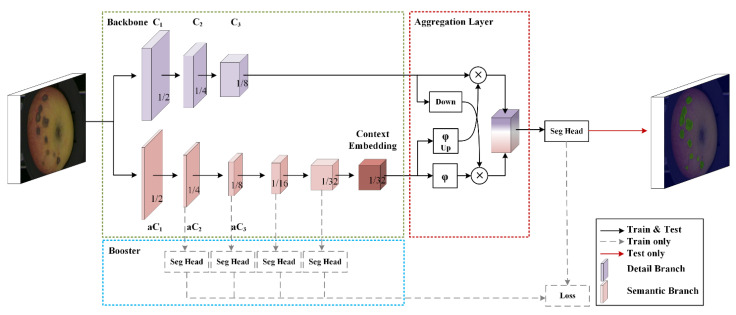
The architecture of BiSeNet V2 network.

**Figure 3 foods-11-03150-f003:**
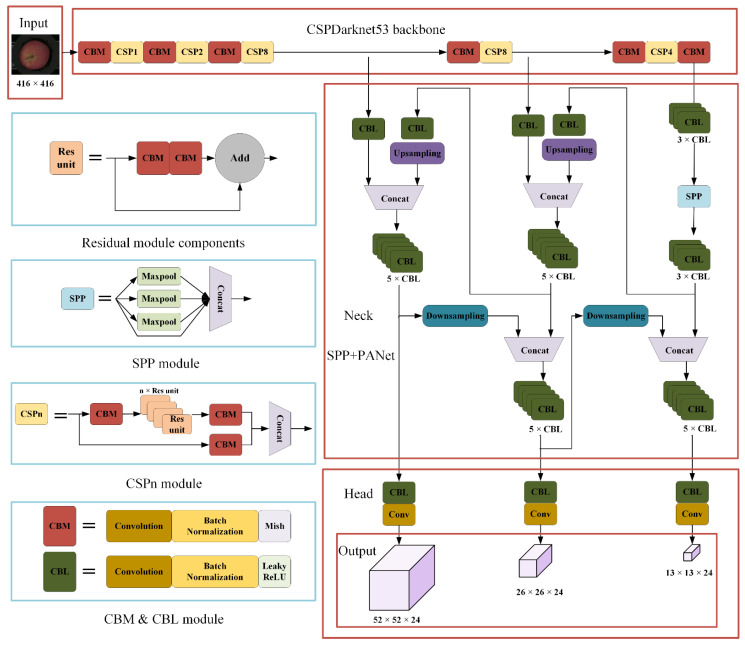
The architecture of YOLO V4 network.

**Figure 4 foods-11-03150-f004:**
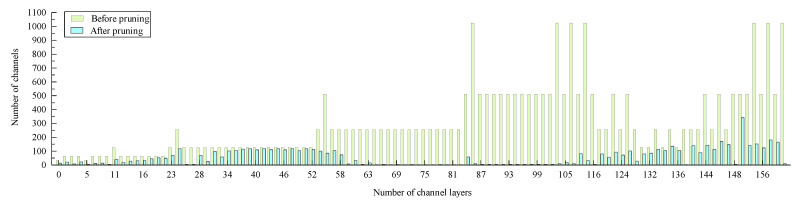
The channel changes of each layer of YOLO V4 model before and after pruning.

**Figure 5 foods-11-03150-f005:**
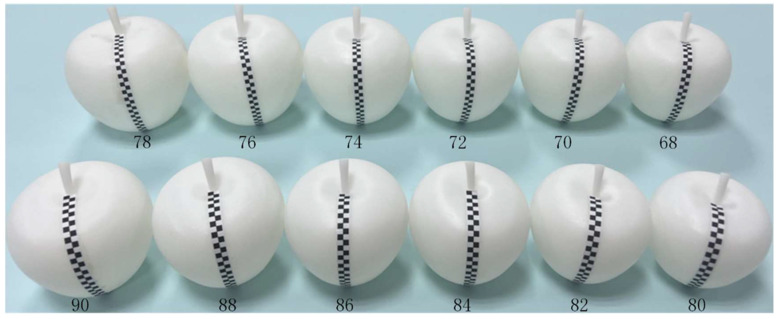
The apple models with labels.

**Figure 6 foods-11-03150-f006:**
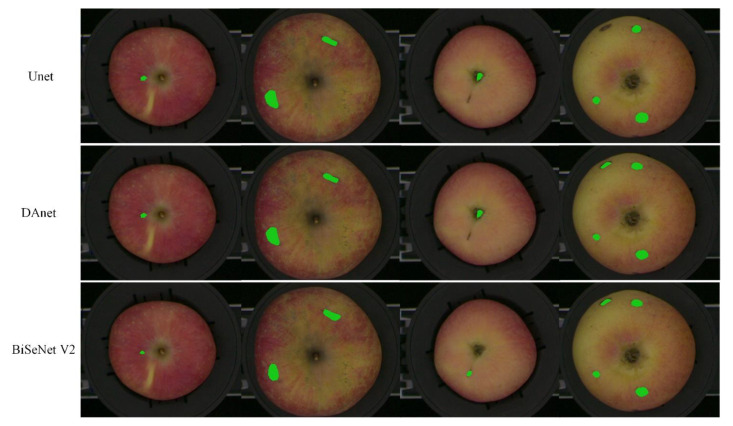
The segmentation results of the Unet, DAnet and BiSeNet V2 networks.

**Figure 7 foods-11-03150-f007:**
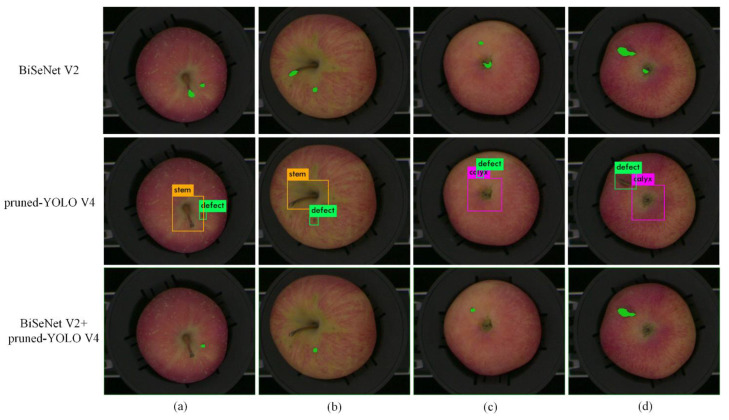
Comparison between segmentation results and detection results.((**a**,**b**) are images taken with stem upward. (**c**,**d**) are images taken with calyx upward.)

**Figure 8 foods-11-03150-f008:**
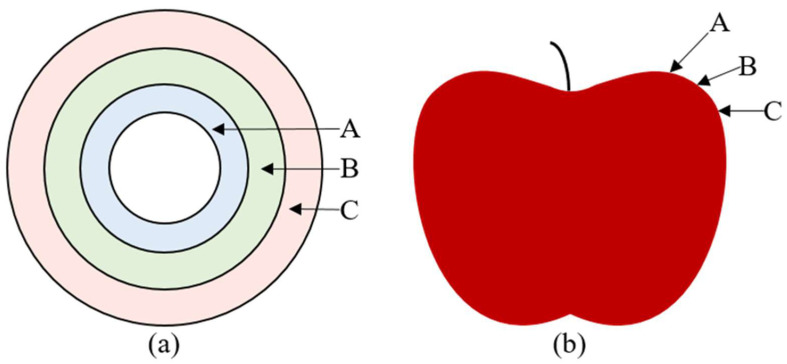
Schematic diagram of defect location. (A, B, C represent different areas of the apple). (**a**) is a diagram of the top view of the apple and (**b**) is a diagram of the front view of the apple) Schematic diagram of defect location. (A, B, C represent different areas of the apple).

**Figure 9 foods-11-03150-f009:**
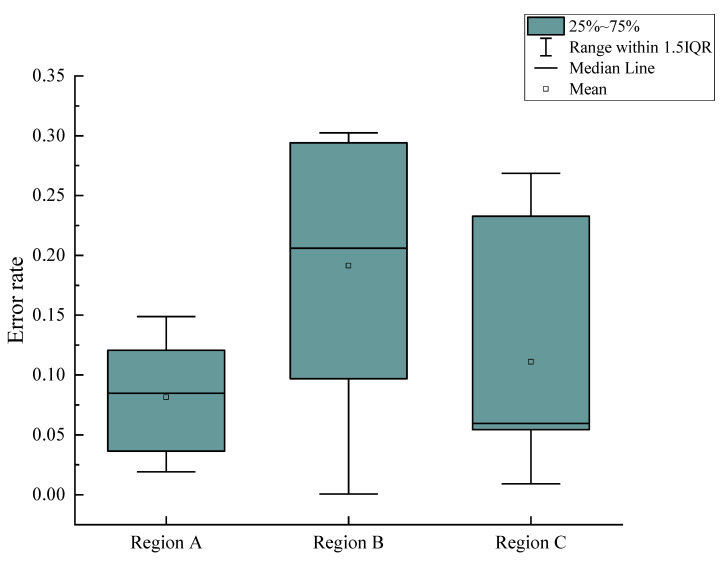
Boxplot of the error rate between calculated and actual defect area at different regions of fruit.

**Figure 10 foods-11-03150-f010:**
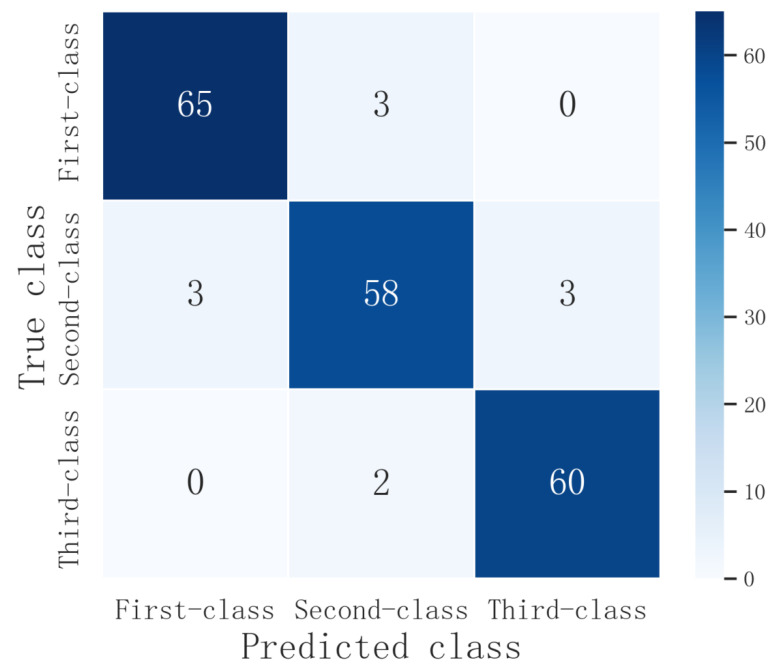
The confusion matrix of defective apple grading.

**Table 1 foods-11-03150-t001:** The initialization parameters of BiSeNet V2 network.

Input Size of Images/Pixel	Batch Size	Initial Learning Rate	Iterations
512 × 512	4	1.1 × 10^−3^	1000

**Table 2 foods-11-03150-t002:** The comparison of different semantic segmentation models.

Models	MIoU/%	MPA/%	Inference Time/ms	Parameters/MB	Model Size/MB
DAnet	74.08	99.52	37.40	45.31	181.30
Unet	73.93	99.47	22.64	12.78	51.15
BiSeNet V2	80.46	99.66	9.00	2.22	9.67

**Table 3 foods-11-03150-t003:** Calculation results of defect area.

Defect Position	The Pixels of the Defect	Calculated Defect Area/mm^2^	Actual Defect Area/mm^2^	Root Mean Square Error
	389.47	39.79	38.48	2.38
A	243.55	24.58	22.90	2.76
	186.92	19.08	18.10	1.85
	370.41	37.75	36.32	1.36
B	218.63	21.86	19.63	2.13
	157.68	15.77	16.62	2.92
	323.90	32.39	33.18	3.16
C	221.58	22.16	20.43	3.22
	135.27	13.53	12.56	3.03

**Table 4 foods-11-03150-t004:** The average of detection results of three grades.

Defect Level	Precision/%	Recall/%	Accuracy/%	F1/%
First class	95.59%	95.59%	-	-
Second class	92.06%	90.63%	-	-
Third class	95.24%	96.77%	-	-
Total	94.30%	94.33%	92.42%	94.31%

## Data Availability

The data presented in this study are available on request from the corresponding author. The data are not publicly available since future studies are related to current data.

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
