# Peer review of "Real-Time Grading of Defect Apples Using Semantic Segmentation Combination with a Pruned YOLO V4 Network"

_foods, 2022, doi:10.3390/foods11193150_

Round 1

Reviewer 1 Report

The present paper entitled “ Real-time grading of defect apple using semantic segmentation combination with pruned-YOLO V4 network” used RGB camera in conjunction with deep learning technique to grade the apples based on the defects. However, the results obtained in the present study were appropriate and technically valid. There are plenty of grammatical mistakes, some of the sentences are not clear and incomplete, authors should seek help from the English native speakers. The authors needs to address or answer the following queries

• Line 14- replace transmit with convey

• Line 15- after bruised add “while image acquisition”

• Abstract flooded with materials and methods parts than results (only 21-23). Abstract needs to improve it should cover all the aspects such introduction, aim, methodology, results and conclusion briefly.

• Line 43- abbreviate ANN

• Introduction too lengthy looking like review of literature

• Line 60-62 sentence is incomplete “non-destructive detection of fruits”, non-destructive detection of what ?

• Line 66-67- MSI requires less time and cost also as compared to HSI.

• Line 68-Abbrivate “PCA”.

• Line 70-71-

• There are lot of papers which are already published on application of image processing techniques to grade and sort the fruits. Justify how your work is different from their work by citing their work.

• The justification part in introduction section is not clear needs improvement.

• HSI had certain advantages as well as disadvantages as compared to RGB imaging but authors failed to discuss clearly about this. Refer the following paper for clear idea

https://doi.org/10.1016/j.tifs.2021.12.021

• Line 88- abbreviate SSD

• Line 104-111 which fruit industries are using roller conveying system, in general fruit industries used belt conveyors to convey fruits and rollers used for grading the fruit based on size.

• Authors should justify their work in terms of image processing and algorithm not in terms of conveying system development.

• Line 127 mention clearly how fruits were washed and any technique to remove the surface water if adopted

• Line 132-133 remove “Schematic diagram as shown in “ just mention Figure 1 in bracket after the light chamber.

• Line 146 provide the specifications of camera.

• Line 148 mention light intensity.

• Figure 1 title need to change.

• The image of fruit was acquired from the top what if damage occurred at the sides of the fruit?

• Line 175 “The above steps were repeated” sentence is not clear for which purpose steps were repeated ?

• Does authors used undamaged fruits for model training ?

• Line 180-182 how fruits were divided into training and validation sets?

• Section 2.4 and 2.5 too much introduction in these sections discuss briefly.

• Line 205 divided “into” three main

• Line 221-224 abbreviate IB, BV, RB

• Table 1. Initial learning rate -3 should be superscript

• Table 1. The input size of image is different from the previously mentioned size (400*336).

• Line 413-416 entire sentence need to reframe, flow is missing.

• Line 426-429 make it single sentence.

• Line 454-455 the figure 9 related to error at different regions not for three apples check the sentence.

• Line 461-464 the entire interpretation need to cross check it justify . The variation in error was due to the defect position as well.

• Line 469 replace scene with “conditions”.

• Figure 9 title change to “Boxplot of the error rate between calculated and actual defect area at different regions of fruit”.

• Table 4 which criteria used to classify the classes .

• Figure 10 in methodology the number of samples used for testing was 600 but here in confusion matrix it showing only 194 ?

• Conclusion part covered more methodology part, it should emphasize more on outcome, scope and future trends. Entire conclusion part need to rewrite.

Reviewer 2 Report

please kindly revise followed my comment in attachment
